# Determination of Aflatoxin M1 in Raw Milk Using an HPLC-FL Method in Comparison with Commercial ELISA Kits—Application in Raw Milk Samples from Various Regions of Greece

**DOI:** 10.3390/vetsci8030046

**Published:** 2021-03-10

**Authors:** Martha Maggira, Maria Ioannidou, Ioannis Sakaridis, Georgios Samouris

**Affiliations:** Department of Hygiene and Technology of Food of Animal Origin, Veterinary Research Institute, Hellenic Agricultural Organization-DEMETER, Campus of Thermi, 57001 Thessaloniki, Greece; marthamaggira@gmail.com (M.M.); ioannidou@vri.gr (M.I.); sakaridis@vri.gr (I.S.)

**Keywords:** aflatoxins, mycotoxins, HPLC, ELISA, milk, aflatoxin M1

## Abstract

The highly toxic Aflatoxin M1 (AFM1) is most often detected in milk using an Enzyme-Linked-Immunosorbent Assay (ELISA) for screening purposes, while High-Performance Liquid Chromatography with Fluorescence Detector (HPLC-FL) is the reference method used for confirmation. The aim of the present study was the comparison between three commercially available ELISA kits and a newly developed HPLC-FL method for the determination of the AFM1 in milk samples. The developed HPLC-FL method was validated for the AFM1 and Aflatoxin M2 (AFM2), determining the accuracy, precision, linearity, decision limit, and detection capability with fairly good results. All three ELISA kits were also validated and showed equally good performance with high recovery rates. Moreover, the Limit Of Detection (LOD) and Limit Of Quantification (LOQ) values were found to be significantly lower than the Maximum Residue Limit (MRL) (50 ng kg^−1^). After the evaluation of all three commercial kits, the ELISA kit with the optimum performance along with the HPLC method was used for the determination of AFM1 in raw cow’s, goat’s, and sheep’s milk samples (396) obtained from producers in different regions of Greece. The evaluation of both methods showed that this ELISA kit could be considered as a faster and equally reliable alternative method to HPLC in routine analysis for the determination of AFM1 in milk.

## 1. Introduction

Aflatoxins (AFs) are one of the most widely known and investigated groups of mycotoxins, which can be found as contaminants in food commodities worldwide [1]. They are toxic substances produced by the secondary metabolism of several fungal species, such as Aspergillus flavus, Aspergillus parasiticus, and Aspergillus nomius, which grow in major food crops under a wide range of climatic conditions [2]. Aflatoxins are chemical substances that cannot be perceived by smell or taste, are fluorescent in ultraviolet light, and are particularly resistant to high temperatures as they withstand exposure above 320 °C without decomposing. Conventional food processing techniques, such as cooking, fermentation, refrigeration, freezing, and pasteurization of food, cannot destroy them [3]. There are four types of aflatoxins produced by A. flavus and A. parasiticus: aflatoxins G1, G2, B1, and B2. Aflatoxin B1 (AFB1) is the most potent natural toxin that is carcinogenic and genotoxic. It is commonly found in many cereal crops and is metabolized by the liver of mammals [4]. Aflatoxin M1 (AFM1) is the major oxidized metabolite of AFB1; it can be found in milk (specifically, it is partitioned into the water and cream parts of it) and other dairy products in which it may be induced by feed carry-over contamination [5]. AFM1 was classified as a possible human carcinogen (Group 1) by the International Agency for Research on Cancer [6]. Exposure to aflatoxins, through consumption of contaminated food and feed, can lead to serious threats to human and animal health by causing various complications, such as hepatotoxicity, teratogenicity, and immunotoxicity [4,7]. The occurrence of AFM1 in milk, especially cow’s milk, makes it a particular risk for humans because of its importance as a foodstuff for adults and especially children [7,8]. For this reason, the European Union and many other countries have set maximum permissible limits for AFM1 in milk (50 ng kg^−1^), cheese (250 ng kg^−1^), and baby food (25 ng kg^−1^) [9,10]. Another aflatoxin that is possible to be found in milk and other dairy products, although less frequently, is the aflatoxin M2 (AFM2), a metabolite of the aflatoxin B2 (AFB2) [11]. However, no limits are set for AFM2 in milk. 

Until recently, thin layer chromatography (TLC) [12], high-performance liquid chromatography (HPLC) coupled with fluorescence detection (FL) [8,13], or mass spectrometry (MS) [14,15,16], were widely used for the analysis of aflatoxins. TLC was the first method used to detect AFM1. Over the last decade, it has been replaced by other chromatographic techniques, such as liquid chromatography (LC) and gas chromatography (GC) coupled with mass spectrometry (MS) [16]. HPLC-FL using post-column derivatization is the reference method used for the qualitative and quantitative determination of mycotoxins [17,18] and is currently the most commonly used method for the determination of AFM1 in milk [13]. However, most of the chromatographic methods employ solid phase pretreatment and immunoaffinity techniques to eliminate interferences and improve the analysis of mycotoxins [19]. Furthermore, the HPLC method requires complicated and time-consuming sample preparation, resulting in the consumption of many chemical solvents [20]. For this reason, at the routine level and in research studies, immunological methods are preferred more than chromatographic, such as Enzyme-Linked-Immunosorbent Assay (ELISA) [21,22]. ELISA is a method that is mainly used in screening control since it gives specific and quick responses with large scale repetition capabilities. It is particularly preferred in routine analysis since it is a low cost, rapid method that requires small sample volumes and fewer preparation procedures than other methods, such as HPLC [23,24]. Although ELISA has the advantage of high specificity and sensitivity, the accuracy of the method depends on the nature of the mycotoxin, the sample preparation process, and the nature of the material, while previous separation allows the accuracy and reproducibility of the method to be improved [25]. However, ELISA could provide false negative or positive results due to the “matrix interference”, which is very common in performing the test, thus contributing to overvaluations or devaluations of the concentration of AFs in the samples [19,26].

The objective of the present study was the comparative evaluation of three commercial ELISA kits for the detection and quantification of aflatoxin M1 in spiked milk samples at known concentrations. The efficacy and reliability of the most efficient ELISA kit were compared with the more sensitive and accurate HPLC technique. This was followed by the testing of unknown raw milk samples (cow, sheep and goat) from different regions of Greece with both analytical methods and a comparative evaluation of the results.

## 2. Materials and Methods

### 2.1. Chemicals and Reagents

Standard solutions of AFM1 (0.5 μg mL^−1^ in acetonitrile) and AFM2 (0.5 μg mL^−1^ in acetonitrile) were purchased from Apollo Scientific LTD (Cheshire, UK). HPLC grade acetonitrile (ACN), methanol (MeOH), and water were obtained from Sigma–Aldrich (Steinheim, Germany). Raw milk was obtained from individual small farms in various regions of Greece. The initial milk sample was prepared in aliquots of 50 mL and was stored at −20 °C until the day of analysis. 

### 2.2. HPLC Instrumentation 

The HPLC system used for the chromatographic determination of aflatoxins residues in milk was the Perkin Elmer Series 200, equipped with a Fluorescence Detector (FL) and 100 μL loop (Perkin-Elmer, Shelton, CT, USA). A vacuum degasser, Perkin Elmer Series 200, directly inserted in the solvent reservoir, achieved degassing of the mobile phase. A mycotoxin C18 5 um, 4.6 × 250 mm, analytical column, purchased from Pickering Laboratory (Mountain View, CA, USA), was used for the separation of examined analytes. The evaluation software was Total Chrom V6.2.0.0.1 with LC instrument control Perkin Elmer.

A glass vacuum filtration apparatus obtained from Alltech Associates (Deerfield, IL, USA) was employed for the filtration of the solvents using cellulose nitrate 0.45 μm membrane filters from Sartorius Stedium Biotech GmbH (Goettingen, Germany) before use. A Vortex Genie 2 (Bohemia, NY, USA), an ultrasonic bath AM-9 Aquasonic Cleaners (Sherwood, AR, USA), and a Hettich Universal centrifugation (Tuttlingen, Germany) were used for the pretreatment of milk samples, whereas immunoaffinity columns (Afla M1) were used for the isolation of AFM1 (Vicam, Milford, MA, USA). All evaporations were performed with a 16-port evaporator model from Barkey GmbH & Co. KG (Leopoldshöhe, Germany). The samples were filtrated with 13 mm × 0.2 μm microfilters, Fioroni (Ingre, France). Moreover, a 20–200 μL micropipette Eppendorf Research (Hamburg, Germany) was used for the preparation of the standard solutions.

### 2.3. ELISA Kits and Instruments Used

Three ELISA kits were obtained and compared for the detection of AFM1 in milk: ELISA BIO SHIELD M1 ES by ProGnosis Biotech A.E. (Larissa, Greece), ELISA AgraQuant Aflatoxin M1 High Sensitivity Assay 5/100 by Romer Labs Singapore Pte Ltd. (Jalan Bukit Merah, Singapore), and ELISA RIDASCREEN Aflatoxin M1 by r-biopharm (Darmstadt, Germany).

Results were evaluated photometrically by the use of the ELISA reader TECAN infinite F50 (Grödig, Austria). The preparation of standard solutions and spiking of milk samples at the appropriate concentrations was achieved by the use of certified pipettes Eppendorf Research plus.

### 2.4. Chromatography

Target analytes were separated by isocratic elution using A: MeOH, B: ACN, and C: water. The initial volume ratio was 22:22:56 (*v*/*v*). The flow rate was set at 1 mL min^−1^ and the time analysis was 10 min. The analytical column was operated at ambient temperature, and the FL detector was set at 365 and 430 nm. The chromatographic conditions are described in the method abstract for post-column liquid chromatography from Pickering Laboratory (Mountain View, CA, USA)

### 2.5. Standard Solution Preparation

For the chromatographic analysis, working standard solutions of each analyte were prepared at a concentration of 50 ng mL^−1^. MeOH:water solution in a 1:1 volume ratio was used as a solvent for the preparation of standard solutions. Aflatoxin solutions were stable for the validation period when stored at −18 °C. Standard solutions were freshly prepared daily by further dilution at various concentrations.

For the analysis using the ELISA kits, a concentrated aqueous solution was prepared at a concentration of 50 ng mL^−1^, from which the standard working solutions were prepared on a daily basis to be used in the validation of the method. The concentrated standard solutions remained at −20 °C for the entire validation period of the method, while the standard working solutions were prepared on a daily basis. All standard solutions remained fully protected from light for the entire duration of their preparation and the spiking of the milk samples to avoid AFM1 inactivation. The standard solutions of known concentrations of AFM1 were used for the preparation of the reference curves, which were provided in the commercial kits of each manufacturer.

### 2.6. Sample Preparation

#### 2.6.1. Sample Preparation Before HPLC-FL Analysis

First, 50 mL of skimmed milk passed through the AflaM1 affinity column, with the aid of a plastic syringe barrel, at a stable rate of 1–2 drops/second until air came through the column. The plastic syringe barrel was removed, and the column headspace was filled with water. The AflaM1 affinity column was washed with 10 mL of purified water at a rate of 2–3 drops/second with the aid of a new syringe barrel. This process was repeated twice. The elution was achieved by passing, first, 1.25 mL of ACN:MeOH (3:2) solution and then 1.25 mL of purified water through the column. The eluent was collected and filtrated with syringe filters before evaporation until 2 mL under a nitrogen stream. Aliquots of 100 μL of the resulting samples were injected into the HPLC system. In the case of fat containing milk samples, centrifugation was applied for fat removal.

#### 2.6.2. Sample Preparation for the ELISA Validation

The milk was transported refrigerated and was thawed one day before the analysis. This was followed by the spiking of milk at known AFM1 concentrations to cover the analysis area at 50–150%. In the present study, the Maximum Residue Limit (MRL) (50 ng kg^−1^) of AFM1 was defined as the analysis area. The milk centrifugation step to separate the supernatant fat layer from the underlying aqueous layer was omitted since skimmed milk was used. The analysis of the samples was followed according to the manufacturers’ instructions. Throughout the analyses, the samples remained fully protected from light to prevent AFM1 inactivation.

## 3. Results and Discussion

### 3.1. Chromatography

The target analytes were separated successfully by isocratic elution. A typical chromatogram is presented in Figure 1. The retention times were observed at 6.58 and 7.796 for AFM2 and AFM1, respectively.

### 3.2. Sample Preparation Before HPLC-FL Analysis

For the sample preparation, the protocol that was reported in the Materials and Methods Section was followed. The proposed sample preparation protocol is simple, non-toxic, with no significant interferences. Absolute recoveries after the procedure ranged from 61–92% for all compounds. Typical chromatograms of a blank and a spiked milk sample are shown in Figure 2a,b, respectively. It is clear that the peaks of the substrate do not interfere with the analytes as they elute at different times.

### 3.3. HPLC-FL Method Validation

The developed HPLC-FL method was validated in terms of linearity, sensitivity, accuracy, precision, decision limit (CCα and detection capability (CCβ), according to European Decision 200/657/EU [27]. Due to the lack of MRL for AFM2, the validation was performed with the MRL level of AFM1. Samples of raw milk, which were analyzed according to the AOAC (Association of Official Analytical Chemists) Method 2000.08 [28] and found not to contain detectable residues of the analytes, were used as blank samples.

Linearity was studied by triplicate analysis of working standard solutions at concentration levels between 0.05 to 5 ng mL^−1^. Standard solutions showed good linearity for the target analytes and high coefficients of determination (0.998 for AFM1 and 0.996 for AFM2). In milk, linearity was examined by triplicate analysis of spiked samples within the range of 0.025–0.5 μg kg^−1^, and calibration curves were calculated. Satisfactory coefficients of determination, 0.999 (AFM1) and 0.996 (AFM2), were achieved over the examined range (Table 1). Limits of detection (LOD) and quantification (LOQ) were considered as the concentration giving a signal-to-noise ratio of 3 and 10, respectively. The observed LODs were lower than the respective MRL value of AFM1 for both analytes. The selectivity of the method was proved by the absence of interference of endogenous compounds in the analysis of blank milk samples.

The precision of the method was based on within-day repeatability and between-day precision. The former was assessed by replicate (*n* = 3) measurements from three spiked milk samples at concentration levels of 25 ng kg^−1^, 50 ng kg^−1^, and 75 ng kg^−1^, which correspond to the 0.5 × MRL, MRL, and 1.5 × MRL of the AFM1, respectively, for both analytes. The relative recovery rates of the analytes added in the spiked samples are presented in Table 1. The between-day precision of the method was established by performing triplicate analysis at the same concentration levels in three days. A triplicate determination of each concentration was conducted for a period of three days.

The decision limit (CCα) was calculated after spiking 20 blank milk samples at MRL of the AFM1. The capability of detection (CCβ) was calculated after the spiking of 20 blank milk samples at the corresponding CCα level of each compound. CCα and CCβ values are presented in Table 1 together with all values derived from the validation procedure for examined parameters.

### 3.4. Validation of the ELISA method and ELISA Kits’ Comparative Evaluation

The plotting of the reference curves was based on the duplicate analysis of the standard solutions supplied in each commercial kit (0, 5, 10, 25, 50, and 100 ng L^−1^ for all three kits; and additionally for kit B only, 250 ng L^−1^) Their equations were based on the absorption values (axis Y) versus the concentrations of the standard solutions (logarithmic axis X). All the reference curves of the kits were logarithmic with coefficients of the determination that ranged from 0.984 to 0.988 (Figure 3). Αll values derived from the validation procedure for the examined parameters are shown in Table 2.

For the repeatability test, four milk samples were spiked with standard AFM1 working solutions at concentrations of 30, 55, 80, and 105 ng kg^−1^. All kits showed satisfactory accuracy and repeatability for the tested concentrations around the MRL range (30, 55, and 80 ng kg^−1^) when the analysis of the spiked samples took place three times within the same day (within-day repeatability). At a higher concentration of 105 ng kg^−1^, only the kits “BioShield ES” and “Ridascreen” gave satisfactory recovery values (83.51% and 82.08%, respectively). However, with the increase in AFM1 concentration in the samples, a clear devaluation of concentrations was observed for all kits. The experimental results, after their statistical processing, are presented in Table 2.

Regarding the analysis of the spiked samples during the three days (between-day precision), nine samples were analyzed, which were spiked with standard AFM1 working solutions at concentrations of 30, 55, and 80 ng kg^−1^, for three consecutive days. The kits “Agraquant” and “BioShield ES” gave satisfactory recoveries and % RSD values in all concentrations. On the contrary, the kit “Ridascreen” gave two comparatively higher recoveries (126.92% and 123.54%) with satisfactory values of % RSD (5.86% and 3.09%) only in the concentrations of 30 and 55 ng kg^−1^.

For the calculation of LOD and LOQ, 20 unspiked milk samples were analyzed, and then the average of the measured concentration and the standard deviation were calculated. All kits showed significantly lower values than the MRL, which all ranged within similar concentrations. More specifically, the lowest detection limit was observed for the kit “BioShield ES” at the concentration of 7.94 ng kg^−1^, and the kits “Agraquant” and “Ridascreen” followed with detection limits at concentrations of 8.04 and 9.44 ng kg^−1^, respectively. As far as the Limit of Quantitation is concerned, this was slightly lower (13.78 ng kg^−1^) for the kit of the company “Agraquant” compared to the other two kits, “BioShield ES” and “Ridascreen” (14.41 and 15.70 ng kg^−1^, respectively).

CCα values were calculated by spiking 20 milk samples with AFM1 standard solutions into the MRL value. Concentration on the permissible limit plus 1.64 times the corresponding standard deviation was equivalent to the decision limit. The CCa values for each commercial ELISA kit are given in Table 2. CCβ values were calculated by spiking 20 milk samples with AFM1 standard solutions into the CCα value. The concentration at the decision threshold plus 1.64 times the corresponding standard deviation was equivalent to the detection capacity. The CCβ values for each commercial ELISA kit are also given in Table 2. All ELISA kits gave comparable results.

According to the results obtained, it may be considered that all the tested kits were comparable regarding AFM1 quantification. However, the kit “BioShield ES” showed slightly better performance. Specifically, it gave the most satisfactory recovery rates in the highest concentration of AFM1 and the lowest LOD and high recovery values in-between day precision, which agrees with the conclusions of a previous study that compared commercial ELISA kits [29]. For the above-mentioned reasons, the kit “BioShield ES” was selected for the comparative study among ELISA and HPLC.

As far as the comparison of ELISA and HPLC is concerned, there were some differences that were observed after the validation of the two methods. Despite being less accurate in the past decades, ELISA has improved significantly over the last few years and is nowadays competitive against other analytical techniques, a conclusion that was also confirmed by our study. Particularly, ELISA demonstrated higher absolute recovery rates. At the same time, LOD, CCα, and CCβ values were found to be lower than the values of the HPLC validation, which comes in agreement with the result of previous studies [21,30]. However, HPLC showed slightly better linearity, precision, and accuracy for the target compound. Another advantage of the HPLC-FL analysis is that it has the ability to detect AFM2 or other mycotoxins simultaneously in the same substrate.

### 3.5. Quality Control of Raw Milk Samples by Means of a Comparative Study among the HPLC-FL Method Developed and ELISA

A total of 396 samples of cow’s (116), sheep’s (228), and goat’s (52) raw milk were examined for the presence of AFM1 residues. The samples were collected from milk-producing facilities in different regions of Greece mentioned in Table 3.

First, the samples were analyzed by the ELISA kit in duplicate according to the manufacturers’ guidance. During the monitoring of the positive milk samples, the ELISA kit determined AFM1 in 39 of 396 milk samples (10.15%), but only three samples contained AFM1 in concentrations above the permissible limit (0.75%). Samples that contained AFM1 in concentrations close to MRL were considered as equivocal negative.

Subsequently, milk samples that were found to be positive and those which were equivocal negative according to the results of the analysis with ELISA were pretreated with the sample preparation protocol described in the Materials and Methods Section and then analyzed by the HPLC-FL method developed herein. Moreover, 54 negative milk samples that were randomly selected (13.6%) were analyzed by the HPLC-FL method for confirmation purposes. From the analysis of negative samples, the latter showed similar results to those obtained by the ELISA method. According to the HPLC analysis, two samples contained AFM1 in concentrations above the permissible limit (0.5%), and this disagreement with the ELISA method is considered to occur from the overestimation of the latter. The results which came from samples analyzed by ELISA demonstrated a satisfactory correlation against the results from the same samples analyzed by HPLC-FL. Moreover, the HPLC method found AFM2 in concentrations above the limit of quantification in five milk samples (1.2%). Our study is in accordance with another study [31] in which ELISA determined AFM1 at concentrations higher than 50 ng kg^−1^ in 6 of 50 samples, while HPLC determined AFM1 in 7 of the latter. The results obtained with ELISA in comparison to those obtained with HPLC for the samples that were found to be positive for AFM1 with concentrations above the MRL are given in Table 4.

Other studies that were carried out in Greece and examined sheep’s and goat’s milk samples presented similar results for the presence of AFM1 in milk. More specifically, in a study monitoring the level of AFM1 in milk produced in the region of Thessaly [32], AFM1 was detected in 43 (18.4%) out of 234 samples analyzed, and only 1.79% of them contained AFM1 above the MRL. Moreover, in another Greek study, only 1.7% [33] of the examined samples were found above the maximum tolerable level. This fact implies that nowadays, there is a decrease in the AFM1 levels in Greece, possibly due to increased farmer awareness and more intensive controls in the dairy industry. Similar results are presented in surveys that were performed in Italy [34,35]; in the first one that was carried out in Sardinia [35], none of the examined samples contained AFM1 above the legal limit, while in the second one, carried out in Sicily, only 3 out of 240 samples were found to contain AFM1 above it. However, another study performed in Egypt and South Africa [31] showed a higher occurrence of AFM1 in milk since 20.45% and 12% of the samples from Egypt and South Africa contained AFM1 at concentrations higher than 50 ng kg^−1^, respectively. This difference could possibly be attributed to the different climatic conditions in these countries that favor aflatoxins production and poor feed storage conditions.

## 4. Conclusions

Nowadays, the most widely used method for the determination of the highly toxic AFM1 for screening purposes is ELISA, while HPLC-FL is still the reference method used for confirmation. The present study shows that the newly developed HPLC-FL method of HPLC was found to be accurate, sensitive, and repeatable, and it is suitable for the determination of AFM1 and AFM2 in raw milk samples. Specifically, the two compounds were successfully isolated from the complex substrate of milk after the appropriate sample preparation and gave high recovery rates.

Concerning the comparative evaluation of the ELISA kits, all kits presented high recoveries for the AFM1 and satisfactory accuracy and repeatability for the tested concentrations. Regarding the LOD and LOQ parameters, all kits showed significantly lower values than the MRL. The kit that showed slightly better performance was selected for the comparative study with HPLC-FL.

Taking into consideration the milk samples’ quality control, the results obtained from the ELISA kit were in agreement with those reported by the HPLC method. In conclusion, it is suggested that ELISA can be considered as a reliable alternative to HPLC-FL, thanks to its advantages (rapid, low-cost, easy to use) and can be used for routine analysis in laboratories for the screening of AFM1 in milk samples.

## Figures and Tables

**Figure 1 vetsci-08-00046-f001:**
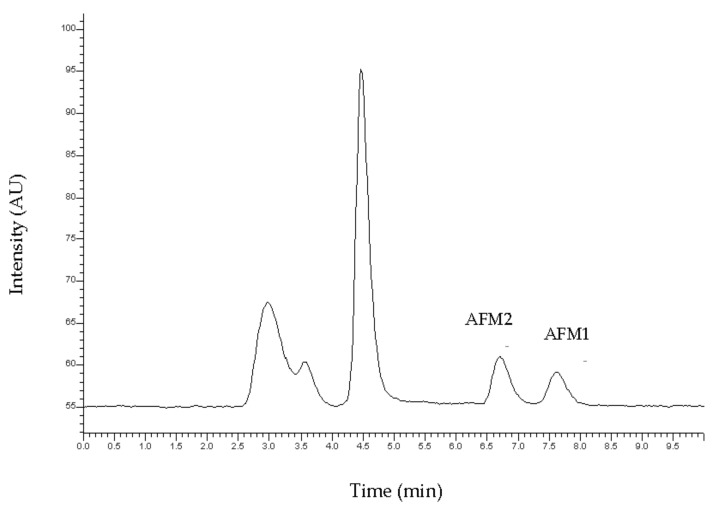
Typical chromatogram of the standard solution of Aflatoxin M1 (AFM1) and Aflatoxin M2 (AFM2) at the concentration of 50 ng kg^−1^.

**Figure 2 vetsci-08-00046-f002:**
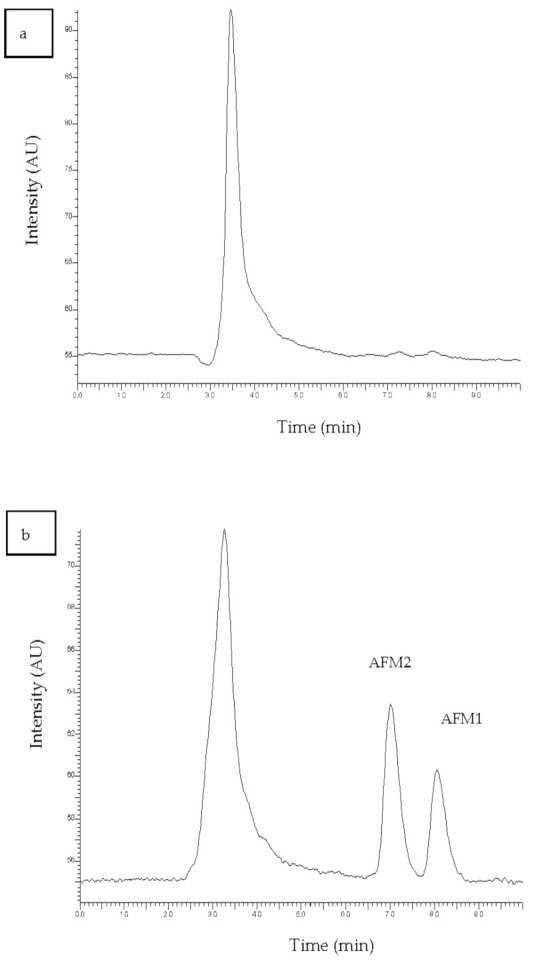
Chromatogram of (**a**) blank milk sample and (**b**) spiked milk sample at a concentration of 50 ng kg^−1^.

**Figure 3 vetsci-08-00046-f003:**
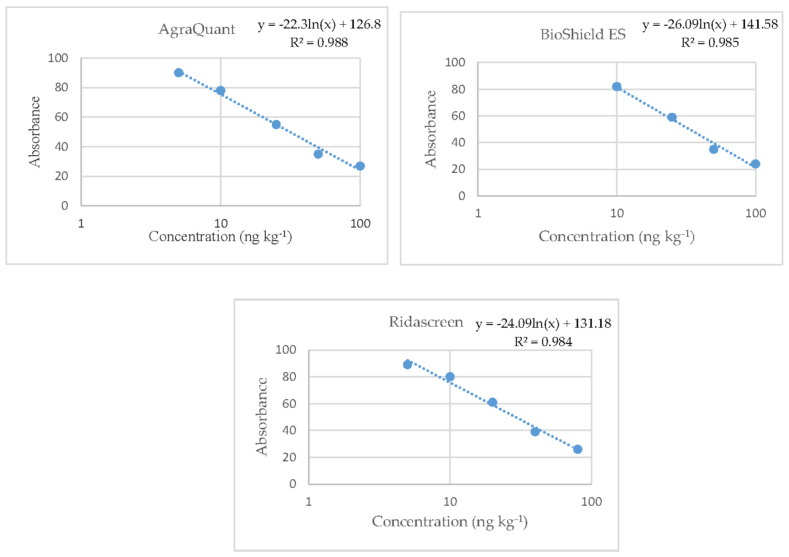
Reference curves of the three Enzyme-Linked-Immunosorbent Assay (ELISA) kits where the straight lines are linear regressions of the absorbance values.

**Table 1 vetsci-08-00046-t001:** Validation parameters for the determination of Aflatoxin M1 (AFM1) and Aflatoxin M2 (AFM2) in milk prior to the High-Performance Liquid Chromatography with Fluorescence Detector (HPLC-FL) method.

Compound	Linearity R^2^	LOD ^1^ (ng kg^−1^)	Intra-Day Recovery (%) RSD ^2^ (%)	Inter-Day Recovery (%) RSD (%)	CCα ^3^ (ng kg^−1^)	CCβ ^4^ (ng kg^−1^)	Error α	Error β	MRL ^5^ (ng kg^−1^)
AFM1	0.999	11.99	90–100 < 3.6	96–109 < 17	56.52	63.97	6.52	6.97	50
AFM2	0.996	16.95	91–119 < 6.1	74–120 < 10.6	57.27	65.57	7.27	8.57	−

^1^ Limit Of Detection; ^2^ Relative Standard Deviation; ^3^ Decision Limit; ^4^ Detection Capability; ^5^ Maximum Residue Limits.

**Table 2 vetsci-08-00046-t002:** Results from the validation of three commercial Enzyme-Linked-Immunosorbent Assay (ELISA) kits of the detection and enumeration of AFM1 in milk samples.

ELISA Kits	Within-Day Repeatability (*n* = 12)			Between-Day Precision (*n = 9*) Recovery (%RSD)	LOD (ng kg^−1^) (X_mean_ + 3*SD) *n* = 20	LOQ (ng kg^−1^) (X_mean_ + 10*SD) *n* = 20	*CCα* (ng kg^−1^)	*CCβ* ng kg^−1^)	Error α	Error β
	AFM1 True Concentration	Precision (%RSD)	Recovery							
Agraquant	AFM1 30 (ng kg^−1^)	6.82%	105%	96−102% (6.33%)	8.04	13.78	53.3	57.3	3.3	4.0
	AFM1 55 (ng kg^−1^)	2.19%	99%	96–113% (13.99%)						
	AFM1 80 (ng kg^−1^)	5.37%	86%	78–86% (13.23%)						
	AFM1 105 (ng kg^−1^)	1.07%	75%	--						
BioShield ES	AFM1 30 (ng kg^−1^)	6.3%	1091%	107–112% (2.37%)	7.94	14.41	53.9	58.1	3.9	4.2
	AFM1 55 (ng kg^−1^)	2.45%	1171%	93–117% (12.65%)						
	AFM1 80 (ng kg^−1^)	2.43%	95%	80–95% (9.97%)						
	AFM1105(ng kg^−1^)	5.47%	84%	--						
Ridascreen	AFM130 (ng kg^−1^)	12.06%	1074%	107–127% (5.86%)	9.44	15.70	54.0	58.8	4.0	4.8
	AFM1 55 (ng kg^−1^)	1.65%	117%	117–124% (3.09%)						
	AFM1 80 (ng kg^−1^)	1.38%	101%	92–105% (12.89%)						
	AFM1 105 (ng kg^−1^)	3.31%	82%	--						

**Table 3 vetsci-08-00046-t003:** Testing of raw milk samples for the presence of AFM1-Distribution by region and type of milk.

Geographical Region	Number of Samples	Cow’s Milk	Goat’s Milk	Sheep’s Milk
Macedonia–Thrace	183	94	25	64
Thessaly	61	20	−	41
Epirus	18	−	8	10
Peloponnese–Ionian Islands	76	−	4	72
Crete	19	2	3	14
Central Greece	39	−	12	27

**Table 4 vetsci-08-00046-t004:** The results from the analysis of the positive milk samples as determined by ELISA and the HPLC-FL method.

Number of Sample	Geographical Region	Type of Milk	ELISA Method (AFM1 ng kg^−1^)	HPLC-FL Method (AFM1 ng kg^−1^)	HPLC-FL Method (AFM2 ng kg^−1^)
37	Macedonia	Cow’s milk	63.3 ± 2.1 > MRL	63.3 ± 2.2 > MRL ^1^	<LOQ
45	Macedonia	Cow’s milk	53.3 ± 1.6 > MRL	47.1 ± 1.9 < MRL	<LOQ
179	Peloponnese	Sheep’s milk	104 ± 3 > MRL	73.4 ± 2.3 > MRL	<LOQ

^1^ Maximum Residue Limits.

## Data Availability

Not applicable.

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
