# Peer review of "Determination of Aflatoxin M1 in Raw Milk Using an HPLC-FL Method in Comparison with Commercial ELISA Kits—Application in Raw Milk Samples from Various Regions of Greece"

_vetsci, 2021, doi:10.3390/vetsci8030046_

Round 1
Reviewer 1 Report
The paper is well structured and the comparison made by the authors is interesting. However, some aspects must be improved before publication.
Specifically:
1. At work it is not clear if authors have simply made a comparison or developed a new HPLC-FL method. If the HPLC-FL method is the result of an optimization carried out by the authors, they must give some detail in the work. If it is a method described in the bibliography, they must indicate the reference.
2. Other sample treatment procedures previously to HPLC-FL determination are described in the literature (for example, Food Chem 333 (2020) 127421). The authors should mention them in the bibliography and compare the analytical parameters obtained with those indicated in previous works. In this way, the authors will be able to highlight the advantages of the proposed method.
3. Authors must take care of the presentation of the results. In Tables 1 and 3, the number of significant figures does not appear to be adequate. For example, there is talk of a recovery of 106.64% -126.92%, does this make sense? Wouldn't it be more appropriate to express it as 107% -127%?
In table 5 the results are not correctly expressed. The confidence interval must be expressed with the correct significant figures. For example, 104.0 ± 3.21 is incorrect, it should be expressed as 104 ± 3.
Author Response
- At work it is not clear if authors have simply made a comparison or developed a new HPLC-FL method. If the HPLC-FL method is the result of an optimization carried out by the authors, they must give some detail in the work. If it is a method described in the bibliography, they must indicate the reference.
Response: Our HPLC-FL method was simply compared with ELISA. We added the details of our method in section 2.4. - Other sample treatment procedures previously to HPLC-FL determination are described in the literature (for example, Food Chem 333 (2020) 127421). The authors should mention them in the bibliography and compare the analytical parameters obtained with those indicated in previous works. In this way, the authors will be able to highlight the advantages of the proposed method.
Response: In Introduction there is discussion for the sample pretreatment and we added this specific reference. - . Authors must take care of the presentation of the results. In Tables 1 and 3, the number of significant figures does not appear to be adequate. For example, there is talk of a recovery of 106.64% -126.92%, does this make sense? Wouldn't it be more appropriate to express it as 107% -127%?
In table 5 the results are not correctly expressed. The confidence interval must be expressed with the correct significant figures. For example, 104.0 ± 3.21 is incorrect, it should be expressed as 104 ± 3.
Response: The values were corrected.
Reviewer 2 Report
See attached document.

Author Response
Overall Comment
More of the abbreviations, to include a larger readership, need to be written out the first time they are used, or an abbreviation reference added to the manuscript.
Response: The abbserviations have be written correctly.
Introduction
Line 43 – These effects can be caused by both long term and short term. Author should state the fraction of milk that contains the most AFM1.
Response: We have changed this point.
Line 59 - FL is only good for mycotoxins that fluoresce, so this statement should be corrected.
Response; AFM1 and AFM2 are mycotoxins that fluorence. We made that point clear
Line 62 – Unclear
Response: We made that point clear.
Line 66 – To be consistent, ELISA should be written out the first time it is used.
Response: We added the abbreviation of ELISA.
Line 88 – The source of the milk should be made clear. Is this milk taken from the bulk milk at small farms or from dairy processers that have collected milk from small farms. Provide the details of how the samples were preserved.
Response: We made clear the source of milk.
Overall comment – Was a standard HPLC-based regulatory method used? If so, it needs to be referenced. If not, what is the source of the HPLC analytical method. Is it an HPLC method that was developed at the Veterinary Research Institute, if so state such. If it is a modified method, the provide the original reference and modifications. For the ELISA kits, were the manufactures recommend methods followed?
What national standard is used for MRL of AFB1?
Response: For the analytical method of HPLC-FL we provided extented details in section 2.4. Moreover we added the method followed for the ELISA in section 2.6.2. MRL of AFB1 is not mentioned in our study.
Results and Discussion
Line 197 – Why was not an internal standard used to determine extraction efficiency?
Response: The external standard was used for the determination of extraction efficiency. We made corrections at line 197.
Round 2
Reviewer 1 Report
In general the authors have answered the questions correctly.
However some errors must be corrected:
- In line 126 MEOH is incorrect, it must be corrected by MeOH
- The tables have lost their formatting and appear mixed with the text.
- In table 3 some recoveries are higher than 100%, it must be corrected (I do not know if it is a problem with the format or an error of the authors when correcting the values)
- In table 5 a result continues to appear incorrectly, 104.0 ± 3.2 must be correctly expressed as 104 ± 3.
- The names of the authors of reference 19 must be corrected, the correct thing is Pellicer-Castell, E .; Belenguer-Sapiña, C .; Amorós, P .; Herrero-Martínez, J. M .; Mauri-Aucejo. A.R
Author Response
-In line 126 MEOH is incorrect, it must be corrected by MeOH
Response: Done.
- The tables have lost their formatting and appear mixed with the text.
Response: Done.
- In table 3 some recoveries are higher than 100%, it must be corrected (I do not know if it is a problem with the format or an error of the authors when correcting the values).
Response: In an analytical method recoveries rates higher than 100% are acceptable.
- In table 5 a result continues to appear incorrectly, 104.0 ± 3.2 must be correctly expressed as 104 ± 3.
Response: Done.
- The names of the authors of reference 19 must be corrected, the correct thing is Pellicer-Castell, E .; Belenguer-Sapiña, C .; Amorós, P .; Herrero-Martínez, J. M .; Mauri-Aucejo. A.R
Response: Done.